# The efficacy and safety of roxadustat for the treatment of anemia in non-dialysis dependent chronic kidney disease patients: An updated systematic review and meta-analysis of randomized clinical trials

Basel Abdelazeem[1,2], Joseph Shehata[3], Kirellos Said Abbas[4], Nahla Ahmed El-Shahat[5], Bilal Malik[1,2], Pramod Savarapu[6], Mostafa Eltobgy[7]*, Arvind Kunadi[1]

1 McLaren Health Care, Flint, Michigan, United States of America, 2 Michigan State University, East Lansing, Michigan, United States of America, 3 Faculty of Medicine, Cairo University, Cairo, Egypt, 4 Faculty of Medicine, Alexandria University, Alexandria, Egypt, 5 Faculty of Medicine for Girls, Al-Azhar University, Cairo, Egypt, 6 Louisiana State University Health Sciences Center, Monroe, Louisiana, United States of America, 7 The Ohio State University, Columbus, Ohio, United States of America

* eltobgy.2@osu.edu

## Abstract

### Background

Roxadustat (ROX) is a new medication for anemia as a complication of chronic kidney disease (CKD). Our meta-analysis aims to evaluate the efficacy and safety of ROX, especially on the cardiovascular risks, for anemia in NDD-CKD patients.

### Methods

Electronic databases were searched systematically from inception to July 2021 to look for randomized control trials (RCTs) that evaluated ROX NDD-CKD patients. Hemoglobin level and iron utilization parameters, including ferritin, serum iron, transferrin saturation (TSAT), total iron-binding capacity (TIBC), transferrin, and hepcidin were analyzed for efficacy. Pooled risk ratios (RRs) and standardized mean differences (SMDs) were calculated and presented with their 95% confidential intervals (CIs).

### Results

Nine RCTs included a total of 3,175 patients in the ROX group and 2,446 patients in the control group. When compared the control group, ROX increased Hb level significantly (SMD: 1.65; 95% CI: 1.08, 2.22; P< 0.00001) and improved iron utilization parameters by decreasing ferritin (SMD: -0.32; 95% CI: -0.51, -0.14; P = 0.0006), TSAT (SMD: -0.19; 95% CI: -0.32, -0.07; P = 0.003), and hepcidin (SMD: -0.74; 95% CI: -1.09, -0.39; P< 0.0001) and increasing TIBC (SMD: 0.99; 95% CI: 0.76, 1.22; P< 0.00001) and transferrin (SMD: 1.20; 95% CI: 0.70, 1.71; P< 0.00001). As for safety, ROX was associated with higher serious adverse effects (RR: 1.07; 95% CI: 1.01, 1.13; P = 0.01), deep venous thrombosis (DVT)

**Data Availability Statement:** All relevant data are within the paper and its Supporting Information files.

**Funding:** The authors received no specific funding for this work.

**Competing interests:** The authors have declared that no competing interests exist.

(RR: 3.80; 95% CI: 1.5, 9.64; P = 0.08), and hypertension (HTN) (RR: 1.37; 95% CI: 1.13, 1.65; P = 0.001).

## Conclusion

We concluded that ROX increased Hb level and improved iron utilization parameters in NDD-CKD patients, but ROX was associated with higher serious adverse effects, especially DVT and HTN. Our results support the use of ROX for NDD-CKD patients with anemia. However, higher-quality RCTs are still needed to ensure its safety and risk of thrombosis.

## Introduction

Anemia is common in non-dialysis chronic kidney disease (ND-CKD) patients and, as the glomerular filtration rate (GFR) decreases, the prevalence increases [1, 2]. The prevalence of anemia in patients with stage 1 CKD is 8.4% compared to 53.4% at stage 5 CKD [3], and it is associated with increased hospitalization, mortality, and a decrease in the quality of life [4, 5]. Therefore, all patients require anemia screening when assessing CKD for the first time and before administering any therapy, and the patient should be monitored frequently for the therapy response [6]. The anemia in CKD patients is multifactorial [7]. Erythropoietin (EPO) is a hormone that stimulates red blood cell synthesis, and it is produced by renal EPO-producing cells located in the interstitium of the outer medulla and cortex [8]. CKD is characterized by interstitial fibrosis leading to EPO insufficiency [9]. Other factors include iron deficiency, decreased oxygen sensing, and accumulation of uremic toxins [10].

The treatment of anemia in ND-CKD patients depends on the anemia severity, and the main treatment options include iron and erythropoiesis-stimulating agents (ESAs) [5], but it is associated with increased risk of serious cardiovascular events, myocardial infarction (MI), stroke, and venous thromboembolism that limit its usage [10–12].

Roxadustat (ROX) is a new medication that reversibly inhibits hypoxia-inducible factor prolyl hydroxylase (HIF-PH) enzymes, leading to improved oxygen sensing and increasing hemoglobin levels. ROX showed efficacy and safety in NDD-CKD patients and had the first global approval in China to treat anemia in NDD-CKD patients [13].

Our study aims to conduct an updated meta-analysis of randomized clinical trials (RCTs) by analyzing previously published RCTs besides the most recent RCTs Akizawa et al. [14], Barratt et al. [15], and Shutov et al. [16]. We will investigate the efficacy and safety of ROX to placebo of ESAs in the treatment of anemia in NDD-CKD patients. Our study will help internists and nephrologists decide whether ROX should be considered in managing anemia in NDD-CKD patients.

## Methods

### Data sources and search strategy

We followed Cochrane Handbook for Systematic Reviews of Interventions and Preferred Reporting for Systematic Review and Meta-Analysis (PRISMA) to conduct this systematic review and meta-analysis [17, 18]. We registered our meta-analysis at OSF Registries with DOI 10.17605/OSF.IO/WGZ6C. Systematic research of PubMed, EMBASE, Scopus, Web of Science, Cochrane Central, and Google Scholar was searched systematically from inception through July 2021 to include citations on non-HD CKD patients treated with ROX for anemia.

The following search terms were used (Roxadustat OR ASP1517 OR FG4592 OR "FG-4592") AND (kidney OR renal) AND (Anemia), and it varies depending on the database (S1 Table). The related articles' feature [19] was used to include any related articles, and the references of the included studies were manually reviewed to include any relevant citation. EndNote [20] was used to save the search result, and the result was transferred to Covidence for screening [21]. A further manual search was done on November 1, 2021, to look for recently published articles.

## Study selection and eligibility criteria

We included studies that met the following criteria: study design was RCTs; written English text; the target population was patients CKD and not on HD; the intervention was ROX and compared placebo or any other medications; primary outcomes changed in hemoglobin level and iron parameters and studies reported the outcomes of interest were included. We excluded the observational studies, studies that did not report a comparator group, and non-randomized studies. Two independent reviewers (JS and KSA) completed the title, abstract, and full-text screening. Any conflict between the authors was resolved by a third author (BA).

## Data extraction

The data from the included studies were extracted independently by three reviewers (JS, KSA, and NAE). The consensus was reached in case of any inconsistency by the fourth author (BA). Each included RCT was abstracted for the first author, published date, country, study design, phase, study period, number, age, gender of patients, and ROX dose.

## Risk of bias assessment

Two Reviewer (KSA and NAE) performed the risk of bias assessment independently using the revised Cochrane risk of bias 2 tool (Rob 2) [22] to evaluate the randomization process, deviations from the intended interventions, missing outcome data, measurement of the outcome, selection of the reported results, and overall risk of bias. The overall grade of each aspect was measured as low risk, high risk, or some concerns.

## Outcomes of interest

The primary outcomes are changes in hemoglobin (Hb) level and iron utilization parameters, including ferritin, serum iron, transferrin saturation (TSAT), Total iron-binding capacity (TIBC), transferrin, and hepcidin. Secondary outcomes are serious adverse events, treatment-emergent adverse effects (TEAEs), and cardiovascular-related adverse effects; hypertension (HTN), hypertensive crisis, pulmonary edema, heart failure, coronary artery diseases, Myocardial infarction (MI), and deep venous thrombosis (DV).

## Statistical analysis

Meta-analyses of all outcomes were performed using RevMan manager v5.3 [23] by Two authors (KSA and NAE) and reviewed by BA. Risk ratios (RRs) were used for the dichotomous outcomes and the standardized mean difference (SMD) for the continuous outcome, and both presented it along with the corresponding 95% confidence interval (CI). Data were analyzed using the Mantel–Haenszel method. A P-value less than 0.05 is considered significant. All outcomes were calculated according to the fixed-effects model (in the absence of significant heterogeneity) and the random-effects model (in the presence of significant heterogeneity). Statistical heterogeneity between trials was evaluated using the Cochran Q test and measured

using $I^2$ statistics. $I^2 > 50\%$ indicates significant heterogeneity across included RCTs [24]. The risk of publication bias was not performed because we included less than ten RCTs [25]. We excluded one study at a time and repeated the analysis to perform the sensitivity analysis by removing one study at a time to assess the impact of each study on the overall study effect on the Hb level. Subgroup analysis was carried out to investigate the impact of the trial phase and the different control groups (ROX Vs. ESA and ROX Vs. placebo) on the Hb level.

We used Comprehensive Meta-Analysis (CMA) [26] software to perform meta-regression to study the effect of rescue therapy or iron supplementation on the mean change of hemoglobin level. Standardized differences in means and the total of ROX versus control were tabulated with the status of rescue therapy and iron supplement; allowed or prohibited. A scatter plot was created for two models: rescue therapy and iron supplement.

## Results

### Search results and study selection

From a pool of 908 potentially relevant articles, we chose 62 for full full-text review according to our inclusion and exclusion criteria. Thus, a total of nine RCTs were included in our systematic review and meta-analysis [14–16, 27–32]. The process of inclusion and exclusion with the reasons for exclusion was shown in the PRISMA flow diagram Fig 1.

### Characteristics of included studies

The main characteristics of the studies included are presented in Table 1. Six RCTs were double-blinded [16, 27–30, 32], one was single-blinded [31], and two were open-label [14, 15]. A total of six RCTs were phase 3 [14–16, 27, 28, 30], and three were phase 2 [29, 31, 32]. Two RCTs compared ROX to darbepoetin alfa [14, 15], and the rest of the RCTs compared ROX to placebo. The ROX dose ranged from 50 mg to 100 mg three times a week, and two studies used the dose according to the weight ranging from 0.7 to 2.25 mg/kg [29, 31]. The study duration ranged from 8 weeks to 104 weeks. The mean age of the included patient was 62 ± 14.8 years, and 43.5% were male. The primary causes of the CKD and the rest of the patients' demographics and baseline characteristics were summarized in Table 2.

### Risk of bias assessment

The overall risk of bias was judged as some concerns for three studies [27–29] and as high risk in six studies [14–16, 30–32]. All studies were judged with some concerns on the randomization process, except two had a low risk of bias [14, 32], and one was a high risk of bias [15]. All studies were judged with a low risk of bias for deviations from intended interventions and measurement of the outcome, except three studies have some concerns [14, 15, 31]. Six studies were judged low risk for the missing outcome data [14, 15, 27–29, 32] and three with some concerns [16, 30, 31]. Finally, all studies were judged with a low risk of bias for selective reporting, except two studies were judged as high risk of bias [31, 32]. Fig 2 demonstrated the risk of bias summary and graph, and the detailed assessment of the risk bias will be found in S1 File.

### Primary endpoints

**Hemoglobin level.** ROX showed an increase in Hb level when compared to control (SMD: 1.65; 95% CI: 1.08, 2.22; P< 0.00001) (Fig 3, Forest plot A). Sensitivity analysis by omitting one study at a time showed a consistent result of ROX on Hb (S2 Table). Subgroup analysis to investigate the effect of the trial phase on the Hb level showed a similar result. Three RCTs were in Phase 2 (SMD: 3.01; 95% CI: 0.91, 5.11; P< 0.005) [29, 31, 32] and six RCTs

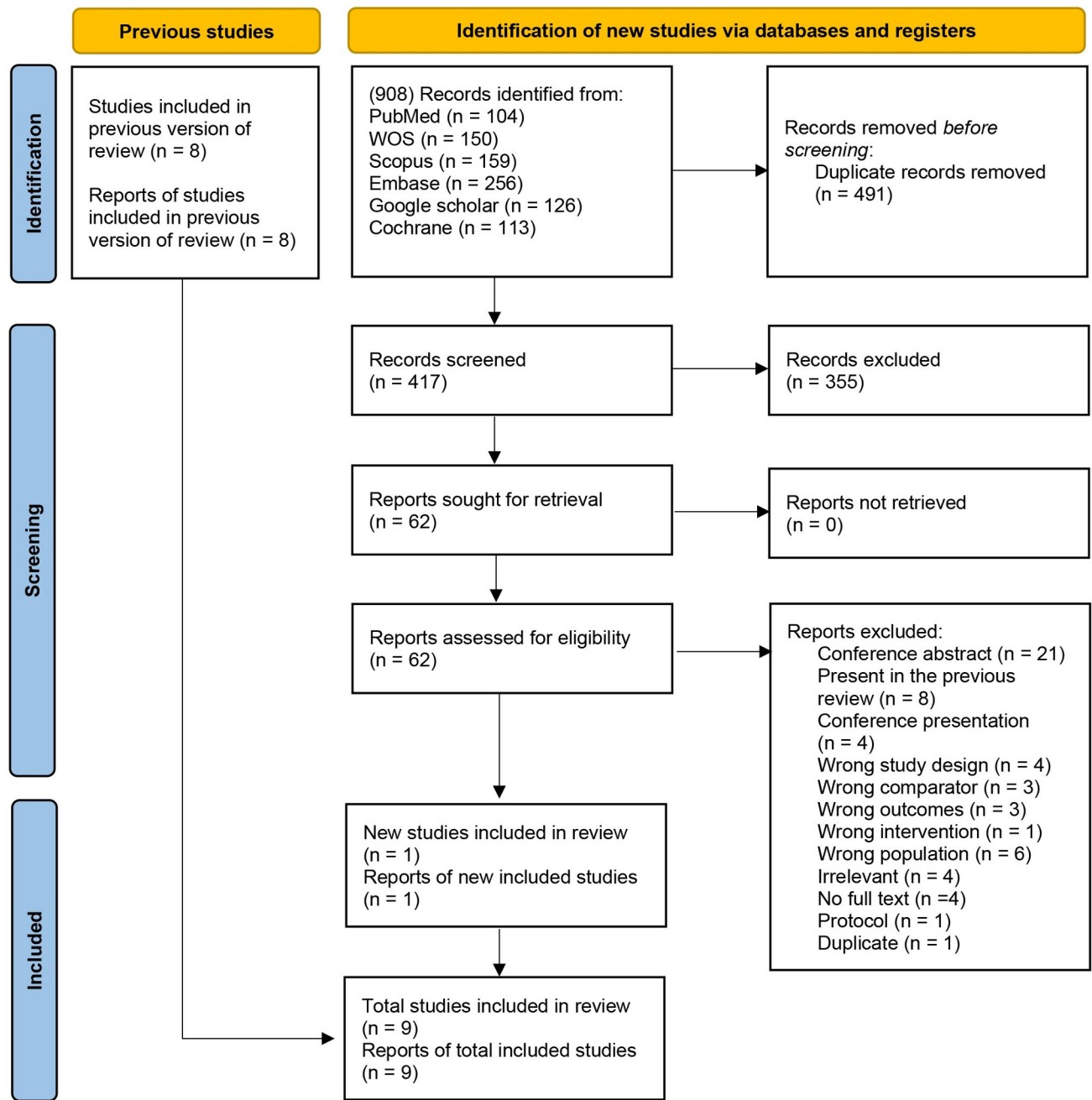

**Fig 1. PRISMA 2020 flow diagram for updated systematic reviews, which included searches of databases, registers, and other sources.**

were in phase three (SMD: 1.12; 95% CI: 0.49, 1.75; P< 0.0005) [14–16, 27, 28, 30] (S1 Fig).
However, when we compared ROX to placebo, there was an increase in Hb level (SMD: 2.11;
95% CI: 1.58, 2.63; P< 0.00001), but when we compared ROX to darbepoetin alfa (DA), there
was no difference between the two groups (SMD: -0.00; 95% CI: -0.14, 0.14; P< 1.00) (S2 Fig).
There were only two RCTs that compared the effect of ROX to DA [14, 15], so more RCTs are
needed to confirm the result.

**Table 1. Characteristics of the included studies.**

| Author, year | Location (sites number) | Study design | Study period | Groups | Number of patients | Age in years, Mean ± SD | Male ratio, % | Roxadustat dose | Study duration | Phase |
|---|---|---|---|---|---|---|---|---|---|---|
| Akizawa et al. 2019 [32] | Japan (32) | Double-blinded RCT | 2013–2015 | Roxadustat | 80 | 64.4 ± 8.7 | 48.80% | (50 mg,70 mg, and 100 mg) TIW | 24 weeks | 2 |
| | | | | Placebo | 27 | 61.9 ± 10.6 | 40.70% | | | |
| Akizawa et al. 2021 [14] | Japan (71) | Open-label RCT | 2017–2020 | Roxadustat | 131 | 68.9 ± 11.6 | 63.40% | (70mg or 100mg) TIW | 52 weeks | 3 |
| | | | | DA | 131 | 70.9 ± 10.2 | 57.30% | | | |
| Barratt et al. 2021 [15] | Europe (200) | Open-label RCT | 2014–2018 | Roxadustat | 323 | 66.8 ± 13.6 | 44.90% | (Weight 45 to 70 kg, 70 mg; weight >70 to 160 kg, 100 mg) TIW | 104 weeks | 3 |
| | | | | DA | 293 | 65.7 ± 14.4 | 44% | | | |
| Besarab et al. 2015 [33] | United States (29) | Single-blind RCT | 2008–2010 | Roxadustat | 88 | 64 | 37.50% | (0.7, 1, 1.5 or 2 mg/kg) BIW or TIW | 4 weeks | 2a |
| | | | | Placebo | 28 | 68.6 | 57.10% | | | |
| Chen et al. 2017 [29] | China (11) | Double-blinded RCT | 2011–2012 | Roxadustat | 61 | 48.9 ± 13.8 | 29.50% | low (1.1–1.75 mg/kg) or high (1.50–2.25 mg/kg) TIW | 8 weeks | 2 |
| | | | | Placebo | 30 | 51.4 ± 11.9 | 26.70% | | | |
| Chen et al. 2019 [30] | China (29) | Double-blinded RCT | 2015–2016 | Roxadustat | 101 | 54.7 ± 13.3 | 36% | (Weight 40 to <60 kg, 70mg; weight ≥60 kg, 100mg) TIW | 8 weeks | 3 |
| | | | | Placebo | 51 | 53.2 ± 13.1 | 39% | | | |
| Coyne et al. 2021 [28] | United States, South America, Australia, New Zealand, and Asia (163) | Double-blinded RCT | 2012–2018 | Roxadustat | 616 | 64.9 ± 12.6 | 39.10% | (Weight 45 to <70 kg, 70mg; weight ≥70 kg, 100mg) TIW | 28–52 weeks | 3 |
| | | | | Placebo | 306 | 64.8 ± 13.2 | 43.50% | | | |
| Fishbane et al. 2021 [34] | 25 Countries (385) | Double-blinded RCT | 2014–2018 | Roxadustat | 1384 | 60.9 ± 14.7 | 40.80% | 70mg TIW | 28–52 weeks | 3 |
| | | | | Placebo | 1377 | 62.4 ± 14.1 | 43.80% | | | |
| Shutov et al. 2021 [16] | Different countries, mainly from Europe (138) | Double-blinded RCT | 2013–2017 | Roxadustat | 391 | 58.25 ± 19.91 | 43.20% | (Weight 45 to ≤70 kg, 70mg; weight >70 to ≤ 160 kg, 100mg) TIW | 52–104 weeks | 3 |
| | | | | Placebo | 203 | 60.5 ± 18.46 | 48.80% | | | |

Continuous variables are expressed in mean ± standard deviation.

RCT = randomized control trial; DA = darbepoetin alfa; TIW = three times a week; DIW = two times a week; SD: standard deviation.

**Iron parameters.** When compared to the control group, ROX showed decrease in ferritin level (SMD: -0.32; 95% CI: -0.51, -0.14; P = 0.0006) (Fig 3, Forest plot B), TSAT (SMD: -0.19; 95% CI: -0.32, -0.07; P = 0.003) (Fig 3, Forest plot C), hepcidin (SMD: -0.74; 95% CI: -1.09, -0.39; P< 0.0001) (Fig 3, Forest plot D). And ROX showed increase in TIBC (SMD: 0.99; 95% CI: 0.76, 1.22; P< 0.00001) (Fig 3, Forest plot E), transferrin (SMD: 1.20; 95% CI: 0.70, 1.71; P< 0.00001) (Fig 3, Forest plot F). There was no difference between ROX and the control group regarding serum iron level (SMD: 0.53; 95% CI: -0.36, 1.42; P = 0.25) (Fig 3, Forest plot G).

## Secondary endpoints

The ROX group showed higher serious adverse effects when compared to the control group (RR: 1.07; 95% CI: 1.01, 1.13; P = 0.01) (Fig 4, Forest plot A). Subgroup analysis comparing ROX to placebo showed a similar result (RR: 1.07; 95% CI: 1.01, 1.13; P = 0.03), However, sub-group analysis comparing ROX to DA showed no difference between the two groups (RR: 1.07; 95% CI: 0.95, 1.21; P = 0.26) (S3 Fig). There was no difference between both groups regarding the TEAEs (RR: 1.02; 95% CI: -1.00, 1.04; P = 0.08) (Fig 4, Forest plot B).

Cardiovascular-related adverse effect; Our results did not show a significant difference in the risk ratio of hypertensive crisis, pulmonary edema, heart failure, coronary artery diseases,

**Table 2. The patient demographics and baseline characteristics.**

| Author, Year | Groups | Race, n (%) | Bodyweight (kg) | eGFRc (mL/min/1.73 m2) | CKD stage |
|---|---|---|---|---|---|
| Akizawa et al. 2019 [32] | Roxadustat | Japanese 80 (100%) | 59.07 ± 9.83 | 16.3 ± 7.8 | 2 to 5 |
|  | Placebo | Japanese 27 (100%) | 60.17 ± 8.72 | 16.3 ± 8.5 |  |
| Akizawa et al. 2021 [14] | Roxadustat | Japanese 131 (100%) | N/A | 17.9 ± 8.2 | N/A |
|  | DA | Japanese 131 (100%) | N/A | 18.2 ± 8.8 |  |
| Barratt et al. 2021 [15] | Roxadustat | White 306 (94.7%), Black 8 (2.5%), Asian 9 (2.8%) | 76.90 ± 16.33 | 20.31 ± 11.49 | 3 to 5 |
|  | DA | White 281 (95.9%), Black 2 (0.7%), Asian 10 (3.4%) | 78.39 ± 17.68 | 20.34 ± 10.73 |  |
| Besarab et al. 2015 [33] | Roxadustat | White 49 (55.7%), Black 34 (38.6%), Asian 2 (2.3%) | N/A | 34.3 ± 12.7 | 3 to 4 |
|  | Placebo | White 15 (53.6%), black 11 (39.3%), Asian 2 (7.1%) | N/A | 31.4 ± 12.4 |  |
| Chen et al. 2017 [29] | Roxadustat | Chinese 61 (100%) | 57.4 ± 11 | 19.4 ± 9.5 | 1 to 4 |
|  | Placebo | Chinese 30 (100%) | 56.9 ±10.3 | 23.0 ±13.4 |  |
| Chen et al. 2019 [30] | Roxadustat | N/A | N/A | 16.5 ± 8 | 3 to 5 |
|  | Placebo | N/A | N/A | 14.5 ± 7.6 |  |
| Coyne et al. 2021 [28] | Roxadustat | N/A | N/A | 21.9 ± 11.5 | 3 to 5 |
|  | Placebo | N/A | N/A | 22.4 ± 11.4 |  |
| Fishbane et al. 2021 [34] | Roxadustat | White 623 (45%), Black 112 (8.1%), Asian 544 (39.3%), American Indian 24 (1.7%) | 69.9 ± 18.5 | 19.7 ± 11.7 | 3 to 5 |
|  | Placebo | White 611 (44.4%), Black 115(8.4%), Asian 538 (39.1%), American Indian 29 (2.1%), Native Hawaiian 2 (0.1%) | 70.6 ± 18.8 | 20 ± 11.7 |  |
| Shutov et al. 2021 [16] | Roxadustat | White 335 (85.7%), Black 10 (2.6%), Asian 9 (2.3%) | 73.86 ± 16.49 | 16.5 ± 10.2 | 3 to 5 |
|  | Placebo | White 182 (89.7%), Black 3 (1.5%) | 76.50 ± 16.51 | 17.2 ± 11.7 |  |

Continuous variables are expressed in mean ± standard deviation.

DA = darbepoetin alfa; eGFR = estimated glomerular filtration rate; CKD = chronic kidney disease; CPN = chronic pyelonephritis; PKD = Polycystic kidney disease; N/A = not applicable.

and MI (S4–S8 Figs). Meanwhile, significant risk was associated with ROX for HTN and (RR: 1.37; 95% CI: 1.13, 1.65; P = 0.001) (Fig 4, Forest plot C) and DVT (RR: 3.80; 95% CI: 1.5, 9.64; P = 0.08) (Fig 4, Forest plot D) respectively.

## Meta-regression

Meta-regression on eight studies showed that resume therapy and iron supplementation led to decrease in the mean difference between the ROX and control group; (SMD: 1.65; 95% CI: 1.08, 2.22; P< 0.00001) before meta-regression versus (SMD: 2.78; 95% CI: .89,4.669; P = 0.009) after regression. A Scatter plot was created for better visualization (S9 and S10 Figs).

## Discussion

In our updated meta-analysis of RCTs, we included 5359 NDD-CKD patients to evaluate the safety and efficacy of ROX compared to placebo or ESA. We concluded that ROX increased Hb level and improved iron utilization parameters by decreasing ferritin, TSAT, hepcidin, and increasing TIBC and transferrin. Those results can be contributed by the effect of ROX on HIF. Hypoxia-inducible factor (HIF) is a transcription factor protein that participates in iron homeostasis and regulates the expression of genes that stimulate erythropoiesis. It consists of an alpha-subunit (HIF- α) and a beta-subunit (HIF- β). HIF- α is oxygen-sensitive and induced by hypoxia, while HIF- β is a constitutive component [7]. Therefore, when the body's oxygen content is normal, the Hypoxia-inducible factor prolyl hydroxylase (HIF-PH) will increase the breakdown of HIF [35]. Conversely, reduced oxygen transport in anemia induces

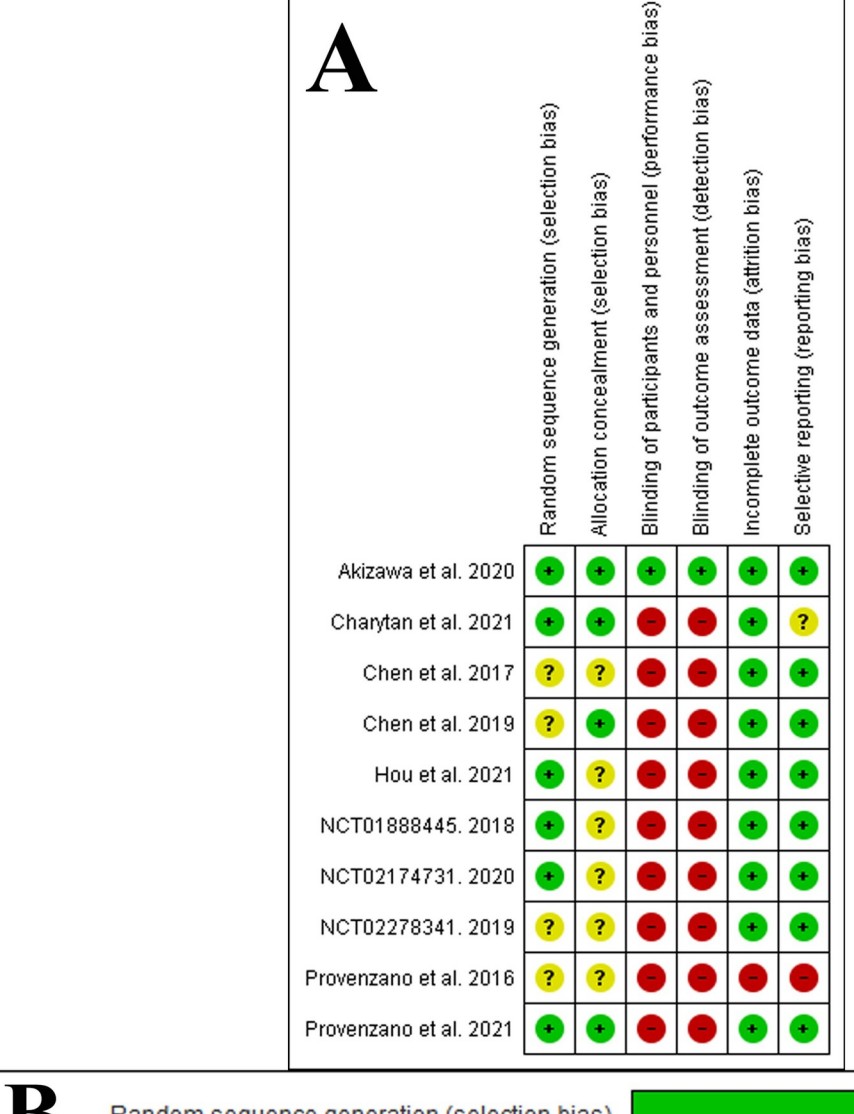

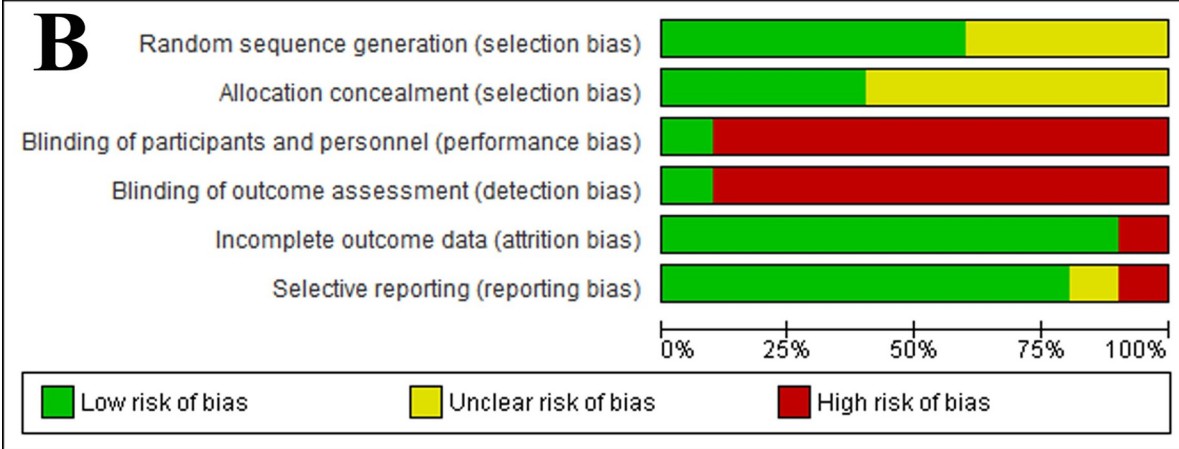

**Fig 2. Risk of bias assessment.** A: Risk of bias summary: review authors' judgments about each risk of bias item for each included study. The items are scored (+) low risk; (-) high risk; (?) some concerns. B Risk of bias graph: review authors' judgments about each risk of bias item presented as percentages across all included studies.

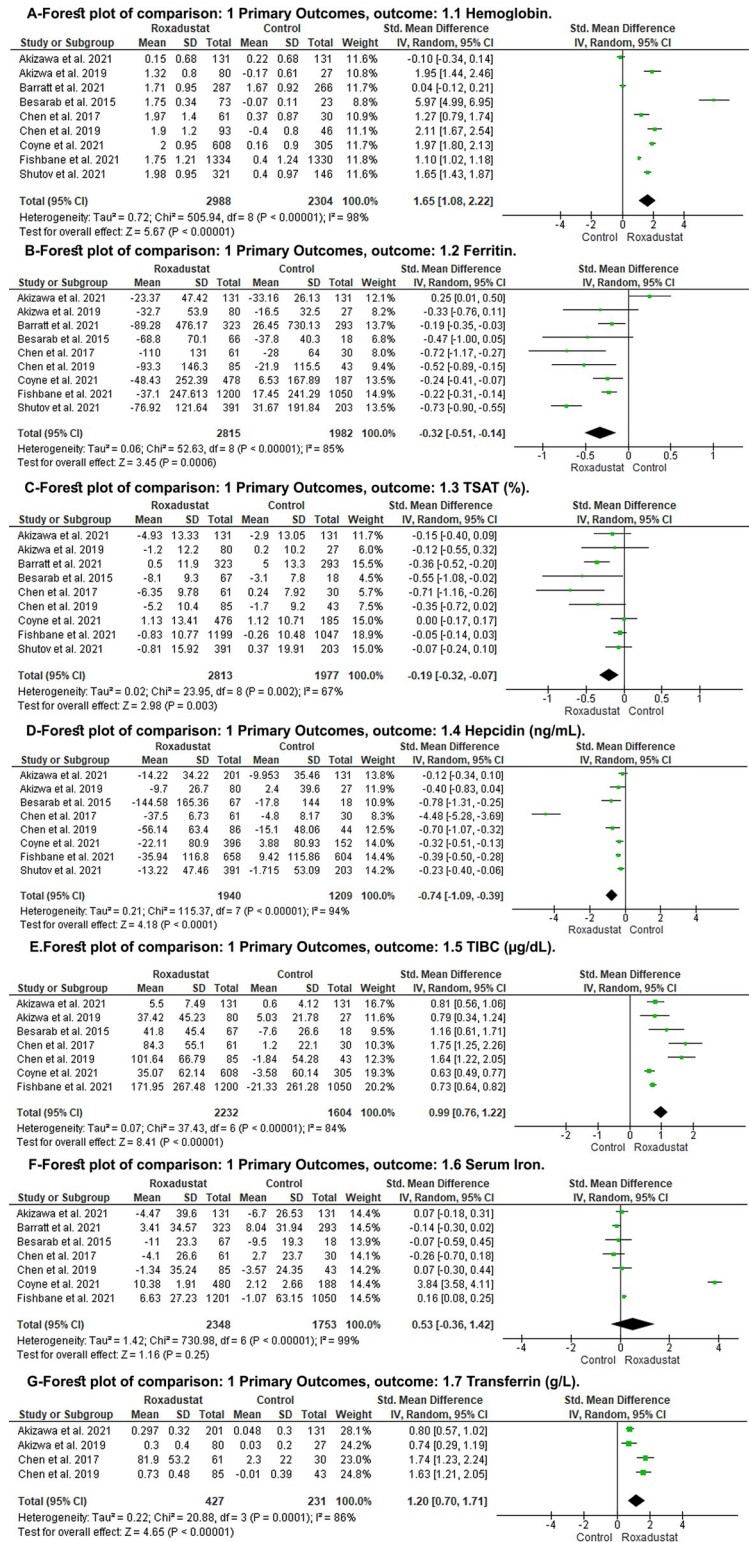

**Fig 3. Forest plot of the primary outcomes.** A: hemoglobin level; B: ferritin; C: TSAT; D: hepcidin; E: TIBS; F: serum iron; G: transferrin; CI: confidence interval; M-H: Mantel-Haenszel; df: degrees of freedom; $I^2$, I-squared.

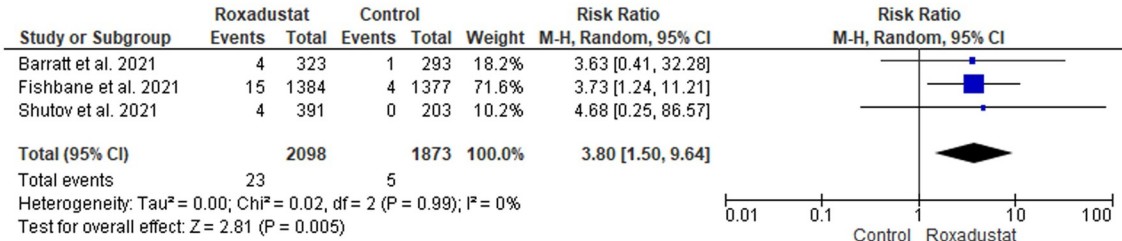

**A-Forest plot of comparison: 3 Secondary outcomes, outcome: 3.1 Serious adverse events.**

| Study or Subgroup | Roxadustat Events | Roxadustat Total | Control Events | Control Total | Weight | Risk Ratio M-H, Fixed, 95% CI |
|---|---|---|---|---|---|---|
| Akizawa et al. 2021 | 23 | 131 | 17 | 131 | 1.4% | 1.35 [0.76, 2.41] |
| Akizwa et al. 2019 | 11 | 80 | 2 | 27 | 0.2% | 1.86 [0.44, 7.85] |
| Barratt et al. 2021 | 209 | 323 | 181 | 293 | 15.2% | 1.05 [0.93, 1.18] |
| Besarab et al. 2015 | 4 | 88 | 1 | 28 | 0.1% | 1.27 [0.15, 10.92] |
| Chen et al. 2017 | 8 | 61 | 4 | 30 | 0.4% | 0.98 [0.32, 3.01] |
| Chen et al. 2019 | 9 | 101 | 6 | 51 | 0.6% | 0.76 [0.29, 2.01] |
| Coyne et al. 2021 | 203 | 611 | 91 | 305 | 9.7% | 1.11 [0.91, 1.37] |
| Fishbane et al. 2021 | 795 | 1384 | 749 | 1377 | 60.2% | 1.06 [0.99, 1.13] |
| Shutov et al. 2021 | 241 | 391 | 115 | 203 | 12.1% | 1.09 [0.94, 1.26] |
| **Total (95% CI)** | | 3170 | | 2445 | 100.0% | 1.07 [1.01, 1.13] |
| Total events | 1503 | | 1166 | | | |

Heterogeneity: Chi² = 2.16, df = 8 (P = 0.98); I² = 0%
Test for overall effect: Z = 2.45 (P = 0.01)

**B-Forest plot of comparison: 3 Secondary outcomes, outcome: 3.2 TEAEs.**

| Study or Subgroup | Roxadustat Events | Roxadustat Total | Control Events | Control Total | Weight | Risk Ratio M-H, Fixed, 95% CI |
|---|---|---|---|---|---|---|
| Akizawa et al. 2021 | 103 | 131 | 92 | 131 | 4.0% | 1.12 [0.97, 1.29] |
| Akizwa et al. 2019 | 63 | 80 | 19 | 27 | 1.2% | 1.12 [0.85, 1.47] |
| Barratt et al. 2021 | 296 | 323 | 271 | 293 | 12.4% | 0.99 [0.95, 1.04] |
| Besarab et al. 2015 | 52 | 88 | 13 | 28 | 0.9% | 1.27 [0.82, 1.96] |
| Chen et al. 2017 | 36 | 61 | 19 | 30 | 1.1% | 0.93 [0.66, 1.31] |
| Chen et al. 2019 | 37 | 101 | 25 | 51 | 1.4% | 0.75 [0.51, 1.09] |
| Coyne et al. 2021 | 564 | 611 | 273 | 305 | 15.8% | 1.03 [0.99, 1.08] |
| Fishbane et al. 2021 | 1243 | 1384 | 1216 | 1377 | 53.0% | 1.02 [0.99, 1.04] |
| Shutov et al. 2021 | 343 | 391 | 176 | 203 | 10.1% | 1.01 [0.95, 1.08] |
| **Total (95% CI)** | | 3170 | | 2445 | 100.0% | 1.02 [1.00, 1.04] |
| Total events | 2737 | | 2104 | | | |

Heterogeneity: Chi² = 7.67, df = 8 (P = 0.47); I² = 0%
Test for overall effect: Z = 1.73 (P = 0.08)

**C-Forest plot of comparison: 4 Cardiovascular side effects, outcome: 4.1 Hypertension.**

| Study or Subgroup | Roxadustat Events | Roxadustat Total | Control Events | Control Total | Weight | Risk Ratio M-H, Random, 95% CI |
|---|---|---|---|---|---|---|
| Akizawa et al. 2021 | 7 | 201 | 5 | 131 | 2.7% | 0.91 [0.30, 2.81] |
| Barratt et al. 2021 | 8 | 323 | 5 | 293 | 2.8% | 1.45 [0.48, 4.39] |
| Chen et al. 2017 | 4 | 61 | 0 | 30 | 0.4% | 4.50 [0.25, 80.95] |
| Chen et al. 2019 | 6 | 101 | 2 | 51 | 1.4% | 1.51 [0.32, 7.24] |
| Coyne et al. 2021 | 95 | 611 | 27 | 305 | 21.3% | 1.76 [1.17, 2.63] |
| Fishbane et al. 2021 | 159 | 1384 | 125 | 1377 | 70.6% | 1.27 [1.01, 1.58] |
| Shutov et al. 2021 | 4 | 391 | 1 | 203 | 0.7% | 2.08 [0.23, 18.46] |
| **Total (95% CI)** | | 3072 | | 2390 | 100.0% | 1.37 [1.13, 1.65] |
| Total events | 283 | | 165 | | | |

Heterogeneity: Tau² = 0.00; Chi² = 3.27, df = 6 (P = 0.77); I² = 0%
Test for overall effect: Z = 3.27 (P = 0.001)

**D.Forest plot of comparison: 4 Cardiovascular side effects, outcome: 4.7 Deep vein thrombosis.**

| Study or Subgroup | Roxadustat Events | Roxadustat Total | Control Events | Control Total | Weight | Risk Ratio M-H, Random, 95% CI |
|---|---|---|---|---|---|---|
| Barratt et al. 2021 | 4 | 323 | 1 | 293 | 18.2% | 3.63 [0.41, 32.28] |
| Fishbane et al. 2021 | 15 | 1384 | 4 | 1377 | 71.6% | 3.73 [1.24, 11.21] |
| Shutov et al. 2021 | 4 | 391 | 0 | 203 | 10.2% | 4.68 [0.25, 86.57] |
| **Total (95% CI)** | | 2098 | | 1873 | 100.0% | 3.80 [1.50, 9.64] |
| Total events | 23 | | 5 | | | |

Heterogeneity: Tau² = 0.00; Chi² = 0.02, df = 2 (P = 0.99); I² = 0%
Test for overall effect: Z = 2.81 (P = 0.005)

**Fig 4. Forest plot of the secondary outcomes.** A: serious adverse effect; B; treatment-emergent adverse effects (TEAE); C: hypertension; D: deep venous thrombosis; CI: confidence interval; M-H: Mantel-Haenszel; df: degrees of freedom; I², I-squared.

tissue hypoxia, which stimulates erythropoietin (EPO) production through activation of the HIF system [36]. Erythropoietin induces erythropoiesis by stimulating the division and differentiation of erythroid progenitor cells by activating several signaling pathways, including Janus kinase /signal transducer and activator of transcription 5, phosphatidylinositol 3-kinase /protein kinase B, and Ras-Raf-MEK-ERK pathways [37].

Roxadustat, also known as FG-4592 or ASP1517, is a HIF- prolyl hydroxylase inhibitor and induces EPO production. ROX is an oral medication with peak concentration achieved in one to three hours, with a mean half-life of 12 to 14 hours [38]. ROX increases the hemoglobin level in a dose-dependent manner. RCTs studies the effect of different doses of ROX; Chen et al. reported that ROX showed a dose-response effect on Hb levels. Hg level increase $\geq$ 1 g/dl from baseline was seen in 80.0% of the patients on low dose regimen (1.1–1.75 mg/kg) compared to 87.1% of the patients with high dose regimen (1.50–2.25 mg/kg) [29]. Akizawa et al. used ROX 50 mg, 70 mg, and 100 mg TIW and reported mean (SD) rate of rise of + 0.200 (5), + 0.453 (5), and + 0.570 (5) for the Hb over the first 6 weeks [32]. Similar findings reported by Besarab et al., the Hb response ranged from 30% in the 0.7 mg/kg two times a week group to 100% in the 2.0 mg/kg two times a week and three times a week groups [31].

ROX improves iron utilization parameters by decreasing ferritin, TSAT, and hepcidin and increasing TIBC and transferrin, thus enhancing the utilization of iron in the body; this may lead to iron deficiency. Therefore, it's advisable to receive proper iron supplementation during the treatment period to avoid any adverse event. ROX was associated with higher serious side effects when compared to placebo. Fishbone et al. reported the most common adverse effect of ROX was end-stage kidney disease, urinary tract infection, pneumonia, and HTN [27]. Our meta-analysis result showed no difference regarding the TEAEs between ROX and placebo or DA. The pooled analysis of global phase III study by FibroGen and AstraZeneca, including 4270 NDD-CKD patients showed comparable risks of a major adverse cardiovascular event (MACE) (HR = 1.08, 95%CI 0.94,1.24), MACE + unstable angina and heart failure requiring hospitalization (HR = 1.04, 95% CI 0.91,1.18) in patients received ROX to the patients received placebo [39]. Those results were consistent with Provenzano et al., who reported that there were no increased risks of MACE (HR, 1.10; 95% CI, 0.96 to 1.27), MACE+ (HR, 1.07; 95% CI, 0.94 to 1.21), and all-cause mortality (HR, 1.08; 95% CI, 0.93 to 1.26) between ROX and placebo groups [40]. Our results showed a significant risk of HTN and DVT in ROX groups which warrant close monitoring. When we compared ROX to DA, there was no difference between the two drugs regarding the serious adverse effect, but only two RCTs were compared ROX to DA in NDD-CKD patients [14, 15].

The previous meta-analysis of RCTs by Tang et al. [41] included eight RCTs with a total of 5,379 NDD-CKD patients. They concluded that ROX was associated with increased hemoglobin level weighted mean difference {WMD}: 1.36 g/dL; 95% CI: 0.90, 1.82; p < 0.00001, transferrin level (WMD: 0.6 g/L; 95% CI: 0.24, 0.95; p < 0.0009), and TIBC level (WMD: 59.90 μg/dL; 95% CI: 38.85, 80.96; <0.00001). In addition, they reported that ROX lowered the hepcidin level (WMD: −51.31 ng/ml; 95% CI: −67.88, −28.12; p <0.00001) and the ferritin and TAST levels in NDD-CKD patients, and these results were almost similar to other previous systematic reviews [42–44]. Also, Tang et al. [41] reported that there is no difference between the treatment-emergent adverse events (TEAEs) of ROX and ESAs or placebo except for serious TEAEs, which was higher in the ROX group (OR: 1.15; 95% CI: 1.02–1.29; p < 0.02). Tang et al. published their article before the release of Barratt et al. [15], limiting their ability to assess and evaluate the article. We used standardized mean differences for outcomes with different units from the RCTs that used different scales to standardize the outcomes. Even though we added one more study [14] in our included studies, our analysis has a higher number of patients and studies in different outcomes showing our more detailed and focused analysis on

NDD-CKD only patients. Liu et al. reported similar results regarding TEAEs as they reported no significant difference among different study groups, but they found an increased risk of developing hyperkalemia in the ROX group [42]. We added an analysis for safety with focusing on cardiovascular adverse effects. We were able to do a detailed meta-analysis with all possible shared outcomes among the included RCTs, detailed sensitivity, subgroup analyses, and meta-regression for concomitant iron supplementations.

The limitation of this study is as follows. First, the overall risk of bias was high in six studies and with some concerns in the other three RCTs, and it is likely due to the lack of a clear randomization process in most of the studies. Therefore, high-quality RCTs are needed in the coming future. Currently, there is an ongoing phase 4 trial assessing the effect ROX versus recombinant human erythropoietin on patients with anemia and CKD (NCT04655027). Second, a high level of heterogeneity detected between the RCTs limits the generalization of our results. The high level of heterogeneity is likely due to different ROX doses and populations. Also, only included RCTs were few in some subgroup analyses, which considered a limitation to our analysis. Third, the included RCTs were sponsored by pharmaceutical companies, and data analysis might be subject to some bias. Fourth, the duration of the included RCTs was short, and they reported only short terms results and could not assess the long-term efficacy and adverse effects of ROX.

In contrast, our study has some strengths. Firstly, we included three RCTs that were recently published. Secondly, we did a detailed meta-analysis and focused on NDD-CKD patients to decrease the potential sources of heterogeneity and bias.

## Conclusions

Our review included nine RCTs to assess the effect of ROX on NDD-CKD patients with anemia. We conclude that ROX was associated with increased Hb level and improved iron utilization parameters by decreasing ferritin, TSAT, and hepcidin and increasing TIBC and transferrin. In addition, ROX was associated with higher serious adverse effects, HTN, DVT when compared to placebo. However, higher-quality RCTs are still needed to confirm the results of our review.

## Supporting information

**S1 Checklist. PRISMA 2020 checklist.** Preferred reporting items for systematic review and meta-analysis.
(DOCX)

**S1 Fig. Forest plot of the effect of trial phase on hemoglobin level.**
(DOCX)

**S2 Fig. Forest plot of the effect of control arm type on hemoglobin level.**
(DOCX)

**S3 Fig. Forest plot of the effect of control arm type on serious adverse effects.**
(DOCX)

**S4 Fig. Forest plot of comparison: 4 Cardiovascular side effects, outcome: 4.2 Hypertensive crisis.**
(DOCX)

**S5 Fig. Forest plot of comparison: 4 Cardiovascular side effects, outcome: 4.3 Pulmonary edema.**
(DOCX)

**S6 Fig. Forest plot of comparison: 4 Cardiovascular side effects, outcome: 4.4 Heart failure.**
(DOCX)

**S7 Fig. Forest plot of comparison: 4 Cardiovascular side effects, outcome: 4.5 Coronary artery disease.**
(DOCX)

**S8 Fig. Forest plot of comparison: 4 Cardiovascular side effects, outcome: 4.6 Myocardial infarction.**
(DOCX)

**S9 Fig. Regression of standardized difference in means on rescue.**
(DOCX)

**S10 Fig. Regression of standardized difference in means on iron.**
(DOCX)

**S1 Table. Search terms and results in different databases.**
(DOCX)

**S2 Table. Meta-analysis of the primary outcomes and sensitivity analysis.**
(DOCX)

**S1 File. The detailed assessment of the risk bias.**
(XLSX)

## Author Contributions

**Conceptualization:** Basel Abdelazeem, Kirellos Said Abbas, Arvind Kunadi.

**Data curation:** Basel Abdelazeem, Joseph Shehata, Kirellos Said Abbas, Nahla Ahmed El-Shahat, Bilal Malik, Pramod Savarapu, Mostafa Eltobgy, Arvind Kunadi.

**Formal analysis:** Basel Abdelazeem, Joseph Shehata, Nahla Ahmed El-Shahat.

**Funding acquisition:** Kirellos Said Abbas.

**Investigation:** Basel Abdelazeem, Joseph Shehata, Kirellos Said Abbas, Nahla Ahmed El-Shahat, Pramod Savarapu, Mostafa Eltobgy.

**Methodology:** Basel Abdelazeem, Joseph Shehata, Kirellos Said Abbas, Nahla Ahmed El-Shahat, Bilal Malik.

**Project administration:** Basel Abdelazeem, Kirellos Said Abbas, Bilal Malik, Pramod Savarapu, Mostafa Eltobgy, Arvind Kunadi.

**Resources:** Basel Abdelazeem, Kirellos Said Abbas, Nahla Ahmed El-Shahat.

**Software:** Basel Abdelazeem, Joseph Shehata, Kirellos Said Abbas.

**Supervision:** Basel Abdelazeem, Kirellos Said Abbas, Pramod Savarapu, Mostafa Eltobgy, Arvind Kunadi.

**Validation:** Basel Abdelazeem, Joseph Shehata, Kirellos Said Abbas, Nahla Ahmed El-Shahat, Bilal Malik, Pramod Savarapu, Mostafa Eltobgy, Arvind Kunadi.

**Visualization:** Basel Abdelazeem, Joseph Shehata, Kirellos Said Abbas, Bilal Malik, Mostafa Eltobgy, Arvind Kunadi.

**Writing – original draft:** Basel Abdelazeem.

**Writing – review & editing:** Basel Abdelazeem, Joseph Shehata, Kirellos Said Abbas, Nahla Ahmed El-Shahat, Bilal Malik, Pramod Savarapu, Mostafa Eltobgy, Arvind Kunadi.

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
