## [Decision Letter · Decision Letter 0]

6 Jan 2022

PONE-D-21-35500The Efficacy and Safety of Roxadustat for the Treatment of Anemia in Non-dialysis Dependent Chronic Kidney Disease Patients: An Updated Systematic Review and Meta-Analysis of Randomized Clinical TrialsPLOS ONE

Dear Dr. Eltobgy,

Thank you for submitting your manuscript to PLOS ONE. After careful consideration, we feel that it has merit but does not fully meet PLOS ONE’s publication criteria as it currently stands. Therefore, we invite you to submit a revised version of the manuscript that addresses the points raised during the review process.

We look forward to receiving your revised manuscript.

Kind regards,

Gianpaolo Reboldi, MD, MSc, PhD

Academic Editor

PLOS ONE

Journal Requirements:

Reviewers' comments:

Reviewer's Responses to Questions

**Comments to the Author**

1. Is the manuscript technically sound, and do the data support the conclusions?

Reviewer #1: Yes

Reviewer #2: Yes

2. Has the statistical analysis been performed appropriately and rigorously? 

Reviewer #1: I Don't Know

Reviewer #2: Yes

3. Have the authors made all data underlying the findings in their manuscript fully available?

Reviewer #1: Yes

Reviewer #2: Yes

4. Is the manuscript presented in an intelligible fashion and written in standard English?

Reviewer #1: Yes

Reviewer #2: No

5. Review Comments to the Author

Reviewer #1: In this work, the Authors investigated by a systematic review and meta-analysis the effects and the safety of Roxadustat (ROX) in CKD patients not on dialysis therapy.

Comments

- Some further meta-analyses were recently published (Wang et al., Ann Palliat Med 2021, 10:4736–46; Qie et al., Int Urol Nephrol 2021, 53:985–97; Zhang et al., Aging 2021, 13:17914–29; Liu et al., Front Med 2021, 8: 724456) and should be added to the article.

- The Authors report in the present work higher serious adverse events, hypertension, and deep venous thrombosis in ROX treated patients. By contrast, a recent meta-analysis (Liu, see above) showed that total adverse events were not significantly different between ROX and placebo as confirmed by trial sequential analysis, with increased risk of hyperkalemia events. A comment is welcomed.

- Three phase 2 trials were included: may this decrease the level of evidence?

- How were managed studies reporting data for more than one ROX dose?

- The Authors report improved iron utilization parameters upon ROX treatment including decreases in ferritin and transferrin saturation and increases in TIBG and transferrin. These changes however also indicate a decrease in body’s iron storage and available iron, and suggest that patients were not receiving appropriate iron therapy for body iron requirement during ROX treatment. It appears that a stranded iron application is necessary in future studies to better characterize the effects of ROX on iron metabolism. Authors would comment on this issue.

Reviewer #2: Abdelazeem and Colleagues, with this systematic review and meta-analysis, tried to extrapolate from the randomized studies available the efficacy and safety of Roxadustat on change in Hemoglobin and related parameters as well as on cardiovascular risk. The study is of interest and statistical analysis is overall of good quality. Moreover, this is an important topic from a clinical research perspective. However, I have several concerns to share with Authors:

- Abstract: I would suggest Authors to change “efficacy and safety of ROX, especially on the cardiovascular system” into “efficacy and safety of ROX, especially on cardiovascular endpoints/risk”. The word system is too general. Moreover, please change “hat evaluated ROX NDD- CKD patients” with “hat evaluated ROX in NDD-CKD patients”;

- English language should be improved throughout the manuscript.

- Introduction is clearly written, and the aim is well depicted in my opinion.

- Methodology is overall accurate in term of study selection and statistical tools used. However, I have a major concern namely the fact that studies were “intervention was ROX compared placebo or any other medications” were included in the meta-analysis. In this way Authors compared for example ROX with darbepoetin or ROX against no ESA. This can introduce a bias if the primary endpoint is the change in Hb levels.

- Please also improve the quality of figures. Most of them are not clearly interpretable in the current form.

- Some subgroup analyses/meta-analyses (for instance panel D of figure 4) included less then 5 studies and this confers low power to statistical finding. Please clarify this in the limitations if possible.

6. PLOS authors have the option to publish the peer review history of their article (what does this mean?). If published, this will include your full peer review and any attached files.

Reviewer #1: No

Reviewer #2: No

---

## [Author Response · Author response to Decision Letter 0]

27 Jan 2022

Mostafa Eltobgy, MD

The Ohio State University

281 W Lane Ave, Columbus, OH 43210

Tel: 551-227-6556

E-mail: eltobgy.2@osu.edu

January 27th, 2021

Dear Editor-in-Chief,

We submit for the revision for our manuscript entitled “The efficacy and safety of roxadustat for the treatment of anemia in non-dialysis dependent chronic kidney disease patients: an updated systematic review and meta-analysis of randomized clinical trials” with ID PONE-D-21-35500 for consideration in your journal. 

Please see the reply to each of the reviewers’ comments below.

Journal Requirements: 

When submitting your revision, we need you to adso much ess these additional requirements.

and

Reply: Thank you so much for your comments. We ensured that our manuscript met PLOS ONE's style requirements

 2. Please include captions for your Supporting Information files at the end of your manuscript, and update any in-text citations to match accordingly. Please see our Supporting Information guidelines for more information: 

http://journals.plos.org/plosone/s/supporting-information

Reply: Thank you so much for your comments. We add captions for the Supporting Information files at the end of the manuscript and update any in-text citations to match accordingly as requested

Reviewer #1: In this work, the Authors investigated by a systematic review and meta-analysis the effects and the safety of Roxadustat (ROX) in CKD patients not on dialysis therapy. 

1. Some further meta-analyses were recently published (Wang et al., Ann Palliat Med 2021, 10:4736–46; Qie et al., Int Urol Nephrol 2021, 53:985–97; Zhang et al., Aging 2021, 13:17914–29; Liu et al., Front Med 2021, 8: 724456) and should be added to the article.

Reply: Thank you so much for your suggestions, we added the suggested articles to our discussion except Wang et al. as it discussed dialysis dependent patients not non-dialysis dependent patients. And our target population is non-dialysis-dependent patients.

2. The Authors report in the present work higher serious adverse events, hypertension, and deep venous thrombosis in ROX treated patients. By contrast, a recent meta-analysis (Liu, see above) showed that total adverse events were not significantly different between ROX and placebo as confirmed by trial sequential analysis, with increased risk of hyperkalemia events. A comment is welcomed. 

Reply: Thank you so much for your comment, we added a comment on Liu et al. article in our discussion and commented about hyperkalemia as suggested.

3. Three phase 2 trials were included: may this decrease the level of evidence?

 Reply: Thanks for your comment. We did a subgroup analysis to investigate the effect of the trial phase on the Hb level. We found it did not affect the result. Please refer to Fig S1 in S1 File.

4. How were managed studies reporting data for more than one ROX dose? 

Reply: Thanks for your comment. The ROX doses were different between studies and we couldn’t do a subgroup analysis because multiple different doses were used in the included RCTs and there was no clear value that we can use to subgroup the ROX doses. We report that in the limitations.

5. The Authors report improved iron utilization parameters upon ROX treatment including decreases in ferritin and transferrin saturation and increases in TIBG and transferrin. These changes however also indicate a decrease in body’s iron storage and available iron, and suggest that patients were not receiving appropriate iron therapy for body iron requirement during ROX treatment. It appears that a stranded iron application is necessary in future studies to better characterize the effects of ROX on iron metabolism. Authors would comment on this issue. 

Reply: Thank you so much for your input, we added this to our discussion

Reviewer #2: Abdelazeem and Colleagues, with this systematic review and meta-analysis, tried to extrapolate from the randomized studies available the efficacy and safety of Roxadustat on change in Hemoglobin and related parameters as well as on cardiovascular risk. The study is of interest and statistical analysis is overall of good quality. Moreover, this is an important topic from a clinical research perspective. However, I have several concerns to share with Authors:

1. Abstract: I would suggest Authors to change “efficacy and safety of ROX, especially on the cardiovascular system” into “efficacy and safety of ROX, especially on cardiovascular endpoints/risk”. The word system is too general. Moreover, please change “hat evaluated ROX NDD- CKD patients” with “hat evaluated ROX in NDD-CKD patients”; 

Reply: Thank you So much for your comments. We edited that in the revision. 

2. English language should be improved throughout the manuscript. 

Reply: Thank you So much for your comments. We revised the whole manuscript grammatically and sent it for language editing by a professional. 

3. Introduction is clearly written, and the aim is well depicted in my opinion. 

Reply: Thank you so much for your comment.

4. Methodology is overall accurate in term of study selection and statistical tools used. However, I have a major concern namely the fact that studies were “intervention was ROX compared placebo or any other medications” were included in the metaanalysis. In this way Authors compared for example ROX with darbepoetin or ROX against no ESA. This can introduce a bias if the primary endpoint is the change in Hb levels. 

Reply: Thank you so much for your comments. Only two studies used darbepoetin and the rest used placebo as the control group. We did a sensitivity analysis by omitting each study (Please refer to Table S2 in S1 File) and this didn’t affect the primary endpoint results. In addition that we did subgroup analysis ROX to placebo and ROX to darbepoetin alfa. We discussed the result in primary endpoint paragraph and the analysis can be seen in Fig S2 in S1 File.

5. Please also improve the quality of the figures. Most of them are not clearly interpretable in their current form. 

Reply: Thank you so much for your comments, We tried to improve the quality. The figures currently are in high-quality TIF format with 300 dpi.

6. Some subgroup analyses/meta-analyses (for instance panel D of figure 4) included less then 5 studies and this confers low power to statistical findings. Please clarify this in the limitations if possible. 

Reply: Thank you so much for your comments, we added this to our limitation as requested.

*********End of review********

Please address all correspondence concerning this manuscript to me at eltobgy.2@osu.edu

Thank you for your consideration of this manuscript. 

Sincerely,

Mostafa Eltobgy, MD

---

## [Decision Letter · Decision Letter 1]

17 Mar 2022

The efficacy and safety of roxadustat for the treatment of anemia in non-dialysis dependent chronic kidney disease patients: An updated systematic review and meta-analysis of randomized clinical trials

PONE-D-21-35500R1

Dear Dr. Eltobgy,

We’re pleased to inform you that your manuscript has been judged scientifically suitable for publication and will be formally accepted for publication once it meets all outstanding technical requirements.

Kind regards,

Gianpaolo Reboldi, MD, MSc, PhD

Academic Editor

PLOS ONE

Additional Editor Comments (optional):

Reviewers' comments:

Reviewer's Responses to Questions

**Comments to the Author**

1. If the authors have adequately addressed your comments raised in a previous round of review and you feel that this manuscript is now acceptable for publication, you may indicate that here to bypass the “Comments to the Author” section, enter your conflict of interest statement in the “Confidential to Editor” section, and submit your "Accept" recommendation.

Reviewer #1: All comments have been addressed

Reviewer #2: All comments have been addressed

2. Is the manuscript technically sound, and do the data support the conclusions?

Reviewer #1: (No Response)

Reviewer #2: Yes

3. Has the statistical analysis been performed appropriately and rigorously? 

Reviewer #1: (No Response)

Reviewer #2: Yes

4. Have the authors made all data underlying the findings in their manuscript fully available?

Reviewer #1: (No Response)

Reviewer #2: Yes

5. Is the manuscript presented in an intelligible fashion and written in standard English?

Reviewer #1: (No Response)

Reviewer #2: Yes

6. Review Comments to the Author

Reviewer #1: (No Response)

Reviewer #2: The article has been improved in term of methodology and presentation of results. I will thus suggest acceptance

7. PLOS authors have the option to publish the peer review history of their article (what does this mean?). If published, this will include your full peer review and any attached files.

Reviewer #1: No

Reviewer #2: **Yes: **Michele Provenzano

---

## [Editor Report · Acceptance letter]

24 Mar 2022

PONE-D-21-35500R1 

The efficacy and safety of roxadustat for the treatment of anemia in non-dialysis dependent chronic kidney disease patients: An updated systematic review and meta-analysis of randomized clinical trials 

Dear Dr. Eltobgy:

I'm pleased to inform you that your manuscript has been deemed suitable for publication in PLOS ONE. Congratulations! Your manuscript is now with our production department. 

Kind regards, 

on behalf of

Prof Gianpaolo Reboldi 

Academic Editor

PLOS ONE